# Cross-cultural adaptation of the 4-Habits Coding Scheme into French to assess physician communication skills

**Alexandre Bellier** [1,2,3,4] *, **Philippe Chaffanjon**[2], **Edward Krupat**[5], **Patrice Francois**[1], **José Labarère**[1,2,3,4]

**1** Quality of Care Unit, Grenoble Alpes University Hospital, Grenoble, France, **2** School of Medicine, Grenoble Alpes University, Grenoble, France, **3** Computational and Mathematical Biology Team, TIMC-IMAG, UMR 5525, CNRS, Grenoble Alpes University, Grenoble, France, **4** CIC 1406, INSERM, Grenoble Alpes University, Grenoble, France, **5** Center for Evaluation, Harvard Medical School, Harvard University, Boston, Massachusetts, United States of America

* abellier@chu-grenoble.fr

## Abstract

### Background

The Four Habits Coding Scheme (4-HCS) is a standardized instrument designed to assess physicians' communication skills from an external rater's perspective, based on video-recorded consultations.

### Objective

To perform the cross-cultural adaptation of the 4-HCS into French and to assess its psychometric properties.

### Methods

The 4-HCS was cross-culturally adapted by conducting forward and backward translations with independent translators, following international guidelines. Four raters rated 200 video-recorded medical student consultations with standardized patients, using the French version of the 4-HCS. We examined the internal consistency, factor structure, construct validity, and reliability of the 4-HCS.

### Results

The mean overall 4-HCS score was 76.44 (standard deviation, 12.34), with no floor or ceiling effects across subscales. The median rating duration of rating was 8 min (range, 4–19). Cronbach's alpha was 0.94 for the overall 4-HCS, ranging from 0.72 to 0.88 across subscales. In confirmatory factor analysis, goodness-of-fit statistics did not corroborate the hypothesized 4-habit structure. Exploratory factor analysis resulted in two dimensions, with the merging of three conceptually related habits into a single dimension and substantial cross-loading for 15 out of 23 items. Median average absolute-agreement intra-class

**Data Availability Statement:** The dataset are available in Harvard Dataverse (view at https://dataverse.harvard.edu/dataset.xhtml?persistentId=doi:10.7910/DVN/IZTLWR).

**Funding:** This study was supported by unrestricted grants from the MACSF Corporate Foundation (19,951, AB: https://www.macsf-exerciceprofessionnel.fr/Publications-actions-et-mecenat/Fondation-MACSF The funders had no role in study design, data collection and analysis, decision to publish, or preparation of the manuscript.

**Competing interests:** The authors have declared that no competing interests exist

correlation coefficient estimates were 0.74 (range, 0.68–0.84) and 0.85 (range, 0.76–0.91) for inter- and intra-rater reliability of habit subscales, respectively.

## Conclusion

The French version of the 4-HCS demonstrates satisfactory internal consistency but requires the use of two independent raters to achieve acceptable reliability. The underlying factor structure of the original US version and cross-cultural adaptations of the 4-HCS deserve further investigation.

## Introduction

Physician communication skills are key components of effective medical consultations [1] and comprise core physician competences that are most desired by patients [2]. Evidence has accumulated, supporting the conclusion that high-quality communication relates with enhanced patient satisfaction [3], greater adherence to treatment [4], better health outcomes [5], and decreased risk of malpractice claims [6]. Many organizations have therefore implemented structured training programs and routinely assessed physicians' communication skills [3, 7].

The Four Habits Coding Scheme (4-HCS) is a standardized instrument designed to assess 23 physician communication skills or behaviors from an external rater's perspective, based on video-recorded consultations [8]. The 4-HCS is based on the conceptual framework of the "Four Habits Model," a training program that was developed within the US Kaiser Permanente Health Maintenance Organization and implemented for teaching effective communication skills to thousands of clinicians in this organization over the two last decades [9]. The Four Habits Model refers to basic medical interview tasks that are organized within four dimensions for didactic purposes, namely, Invest in the beginning (six items), Elicit the patient's perspective (three items), Demonstrate empathy (four items), and Invest in the end (ten items) [9].

The original US version of the 4-HCS demonstrated acceptable inter-rater reliability and evidence for construct validity despite moderate internal consistency, with Cronbach's alpha coefficients ranging from 0.51 to 0.81 across the Four Habits, over 100 video-recorded physician–patient encounters [8]. Since its original development, the 4-HCS has been utilized outside the Kaiser Permanente system [9–13]. Cross-cultural adaptations of the 4-HCS have been published in different languages: Norwegian [14], German [15], and Brazilian Portuguese [16]. Given the potential of the 4-HCS for assessing baseline communication skills and measuring the effectiveness of a training program aiming to alter communication skills [9], there is a need for a French version that can be used with medical students during the 4-year competency-based communication curriculum. To our knowledge, only three studies examined the psychometric properties of the 4-HCS and none has investigated the underlying factor structure [8, 14, 15]. Although the findings of previous studies were promising, the authors recommended examining the validity and reliability of the 4-HCS further in different settings and populations [8].

In the present study, we aimed to perform the cross-cultural adaptation of the 4-HCS into French and to assess the psychometric properties of the adapted version, using the original data of video-recorded medical student consultations with standardized patients.

## Materials and methods

### Study design

A two-step procedure was used. Firstly, the 4-HCS was translated and cross-culturally adapted into French. Secondly, the psychometric properties of the French version were studied, with regard to internal consistency, validity, and reliability. The present study was conducted in the Grenoble Alpes University Hospital and the University Grenoble Alpes School of Medicine, France.

### Cross-cultural adaptation of the 4-HCS into French

The translation and cross-cultural adaptation of the source version of the 4-HCS and its code-book were performed by Mapi Language Services, according to published guidelines [17]. Mapi Language Services is an international company with expertise in the field of translation and cross-cultural adaptation of patient reported outcome measures (www.mapigroup.com/services/language-services/).

   The aim of the cross-cultural adaptation process was to obtain a French translation that was conceptually equivalent to the US source version, culturally relevant to the French context, and easily understood by the people who would use the instrument. For this purpose, we used a rigorous methodology involving input from the 4-HCS developer on conceptual issues and a centralized review process coordinated by a consultant with experience in the field. This process included a common understanding of the 4-HCS concepts by all participants involved in the project, quality control by translators, and discussion about translation decisions at each step.

   Practically speaking, the 4-HCS developer (EK) was contacted to obtain permission to use and translate the instrument and to invite him to participate in the project. Two qualified native French-speaking translators independently translated the source version of the 4-HCS into French. A single version was obtained after a reconciliation meeting of the two translators. Then this version was back-translated into English by a third qualified native English speaker, who was blinded to the original US version. The back translation was reviewed for semantic and operational equivalence against the source version of the 4-HCS. We followed a universal-ist approach for equivalence, assessing conceptual, item, semantic, operational, measurement and functional equivalence [18]. After the resulting version was pretested by two raters using 63 video-recorded medical student consultations with standardized patients, minor adjust-ments were made to obtain the final version. The principal investigator checked the proofs of the final version and corrected any errors.

### 4-HCS scoring

In accordance with the source version of the 4-HCS, each item was rated on a 5-point Likert scale, ranging from 1 to 5, with higher scores denoting better performance. The midpoint (i.e., 3) and the two endpoints (i.e., 1 and 5) were anchored, with specific behavioral descriptions [8]. The raters were encouraged to use the midpoint and endpoint categories, with other cate-gories (i.e., 2 and 4) to be used only if they thought that communication skills fell between the anchored points [8]. This approach ensured full use of the 5-point Likert scale [8].

   An overall communication skill score was computed by summing ratings for the individual items, ranging from 23 (i.e., less effective) to 115 (i.e., more effective). Four subscale scores were also computed, each of them corresponding to a key dimension of communication skills (i.e., Invest in the beginning [range, 6–30], Elicit the patient's perspective [range, 3–15], Dem-onstrate empathy [range, 4–20], and Invest in the end [range, 10–50]).

## Physician–patient relationship competence assessment scale

The physician–patient relationship competence assessment scale was developed in French and validated in Canada [19]. This scale consists in 15 items exploring two dimensions of interpersonal skills, namely "understanding of the patient's disease experience" (eight items) and "efficient and respectful communication" (seven items). Each item could be rated on a 4-point Likert scale, ranging from 15 (i.e., less effective) to 60 (i.e., more effective).

## Study sample and data collection

The study sample consisted of video-recorded medical student consultations with standardized patients. All 218 medical students were invited to participate as part of the 4-year competency-based communication curriculum. They were allocated in consecutive alphabetical order based on their surname to the 1st or 2nd semester sessions that took place in October 2017 and April 2018, respectively.

Standardized patients were 20 actors recruited at the Department of Performing Arts in Grenoble Alpes University. Seven standardized medical consultation scenarios were developed and combined with nine personality types or character traits. The actors were instructed on each medical consultation scenario by two clinicians during 2-h sessions. Then they were trained by their improvisation instructor on each personality type or character trait. The combinations of medical consultation scenarios with personality type or character trait were pretested as part of a pilot study.

All medical consultations with standardized patients were video-recorded using professional video equipment. Video-recording of consultations made it possible to overcome some of the challenges of direct observation [20]. Video-recording accurately recorded all events that occurred during consultations, allowing raters to verify their observations as many times as necessary [20]. Video-recordings could be rated by different observers without consultations being disrupted [21]. Finally, video-recording allowed for providing medical students with feedback on their own performance [20, 22].

Four raters were recruited for the project, including a full professor of medicine (PC) and a resident in medicine (AB), both with experience in teaching communication skills, and two medical students. They underwent a training session, which consisted in independently coding five video-recorded consultations with the 4-HCS and subsequently discussing these ratings. After completing the training, the raters independently rated consultations, with each consultation being rated by at least three different raters. Additionally, two of the raters evaluated each video-recorded medical student consultation twice, with the two ratings 2 months apart, in order to quantify intra-rater reliability.

Data for both the cross-cultural adaptation of the 4-HCS and the physician–patient relationship competence assessment scale were captured using an electronic case report form. The completion of each item was mandatory, so there were no missing values.

## Sample size

A sample size of 200 video-recorded consultations was required for confirmatory factor analysis, based on previous simulations [23]. Assuming a Cronbach alpha coefficient point estimate close to 0.80, this sample size would provide a precision of ±0.07 (i.e., 95% confidence interval [CI] ranging from 0.73 to 0.87).

Assuming an intra-rater correlation coefficient point estimate of 0.80, we estimated that a sample of 117 video-recorded consultations would achieve 80% power to demonstrate that it would be higher than 0.70, with a 0.05 two-sided significance level [24]. Assuming an inter-rater correlation coefficient point estimate of 0.80 with four raters, we estimated that a sample

of 68 video-recorded consultations would achieve 80% power to demonstrate that it would be higher than 0.70, with a 0.05 two-sided significance level [24].

## Statistical analysis

**Descriptive statistics.**   To account for inter-rater variability, we computed the students' average 4-HCS overall and subscale rating scores [25]. The 4-HCS overall and subscale scores were reported as means along with standard deviations. The numbers and percentages of responses on anchor points for items and overall and subscale scores were examined to detect floor or ceiling effects. Floor and ceiling effects lower than 15% for subscale scores were considered acceptable [26].

**Internal consistency.**   Internal consistency was evaluated through average inter-item correlation, item-rest correlation (i.e., the correlation between an item and the score that was formed by all other items in the subscale), and Cronbach's alpha coefficient [27]. The internal consistency criterion was fulfilled for item-rest correlation >0.40, and Cronbach's alpha >0.70 was considered satisfactory [26].

**Internal structure.**   The internal structure of the cross-cultural adaptation of the 4-HCS into French was verified using confirmatory and exploratory factor analysis, following current guidelines [27]. First, structural equation modeling of the four predefined habits was carried out with the 23 items assigned to the intended habits to determine whether the video-recorded medical consultation data fit with the internal structure of the original US version of the 4-HCS. This structural equation modeling corresponded to an external model representing the relationships between latent variables (i.e., the four predefined habits) and the manifest variables (i.e., the 23 related items). Various goodness-of-fit statistics were obtained, including the comparative fit index (CFI) [28], the standardized root mean squared residual (SRMR), and the root mean square error of approximation (RMSEA) along with its 90% CI [29]. A CFI value of 0.90 or higher and a SRMR value lower than 0.08 were considered indicative of satisfactory model fit. A 90% CI lower bound for the RMSEA estimate lower than 0.05 would not reject the hypothesis that the fit was close. An upper bound higher than 0.10 would not reject the hypothesis that the fit was poor.

Second, exploratory factor analysis was performed in order to examine possible alternative structures to the original US version of the 4-HCS. An orthogonal rotation method (Varimax) of factors with eigenvalues higher than 1.00 was used, assuming that they were independent [30]. Primary loadings on intended dimensions higher than 0.40 with cross-loadings lower than 0.30 were considered satisfactory.

**Construct validity.**   We assessed construct validity by comparing 4-HCS overall and subscale score values between first and second-semester video-recorded medical consultations. We hypothesized that mean scores were higher for medical consultations recorded during the second semester. Indeed, second semester students were assumed to be more experienced in basic medical interview tasks and more sensitized to communication skills than their counterparts evaluated during the first semester. We also examined convergent validity between the 4-HCS and the physician–patient relationship competence assessment scale, using Pearson correlation coefficients.

**Reliability.**   Inter- and intra-rater reliability assessment complied with the *Guidelines for Reporting Reliability and Agreement Studies* (GRAAS) [31]. The reliability of the French version of the 4-HCS overall and subscale scores was quantified by the intra-class correlation coefficient (ICC) [32]. The ICC is suitable for reliability studies with unbalanced designs involving different sets of raters [33]. Both absolute- and consistency-of-agreement ICCs were computed. Under the absolute-agreement approach, the ratings were considered in absolute

agreement if the 4-HCS scores from all raters matched exactly [32]. Under the consistency-of-agreement approach, the ratings were considered consistent if the 4-HCS scores from any two raters differed by the same constant value for all video-recorded consultations. This implied that raters gave the same ranking to all video-recorded consultations [34]. Individual and average ICCs were estimated, with average ICCs computed over two raters. Although the agreement measured between individual ratings is more common, the use of average ICCs was indicated in this study because the 4-HCS was intended to be used by teams of raters for assessing video-recorded medical consultations. An ICC value equal to or higher than 0.70 was indicative of satisfactory reliability [14].

The study protocol was approved by the Comité d'Ethique du Centre d'Investigation Clinique de Clermont-Ferrand, France (IRB 5891). All participants received information from the principal investigator about the study's overall purpose and the confidentiality requirements and they then provided written informed consent.

## Results

Mapi Language Services translated the 4-HCS scale in January 2018 and issued a translation validation certificate on February 19, 2018. They scrupulously complied with the protocol drawn up according to international standards, producing a cross-cultural adaptation of the 4-HCS scale in line with expectations. The French version of the 4-HCS is shown in S1 Appendix. The full version included a translation of the "codebook," a detailed scoring guide for each item with examples of behaviors and suggested ratings.

Of 218 eligible medical students, 200 (92%) participated in the study. A total of 200 consultations with standardized patients were video-recorded, including 115 and 85 during the first and second semesters, respectively (Fig 1). The median duration was 8 min (range, 4–19 min) per video-recorded consultation. Inter-rater reliability assessment involved 800 ratings while 400 ratings contributed to intra-rater reliability assessment (Fig 1).

The mean 4-HCS score was 76.44 (standard deviation, 12.34) for 200 video-recorded medical consultations, with no floor or ceiling effects observed for subscales (Table 1). Yet the highest (5/5) and lowest values (1/5) accrued more than 15% of the respondents for three items and one item, respectively. Mean 4-HCS scores ranged from 66.97 (SD, 10.26) based on 200 video-recorded consultations for rater 1 (i.e., the least experienced rater) to 93.42 (SD 13.46) based on 85 video-recorded consultations for rater 4 (i.e., the most experienced rater).

Cronbach's alpha was 0.94 (95% CI, 0.93–0.95) for the 4-HCS, ranging from 0.72 to 0.88 across habit subscales (Table 1). All but one item fulfilled the internal consistency criterion, with item-rest correlations higher than 0.40. The exception was item 16 with the item-rest correlation as low as 0.15. Removing this item from the "Invest in the end" habit subscale improved Cronbach's alpha from 0.88 to 0.90.

In confirmatory factor analysis, structural equation modeling of 23 items apportioned in four latent factors yielded CFI (0.79) and SRMR (0.09) estimates that did not achieve recommended thresholds (CFI >0.90 and SRMR <0.08, respectively). The RMSEA estimate was 0.12 (90% CI, 0.10–0.13), with the 90% CI lower bound not rejecting the hypothesis that the fit was close and the 90% CI upper bound not rejecting the hypothesis that the fit was poor.

Exploratory factor analysis of the 23 items identified four principal components with eigenvalues higher than 1.0 and explaining 66.8% of overall variance (Table 2). Graphical assessment of the scree plot suggested that the instrument was close to unidimensionality (S2 Appendix), supporting the use of an overall 4-HCS score. Yet, the factorial structure for the French version of the 4-HCS departed from the hypothesized four-dimension structure. Twenty-one out of 23 items had primary factor loadings over 0.40 while 15 items yielded

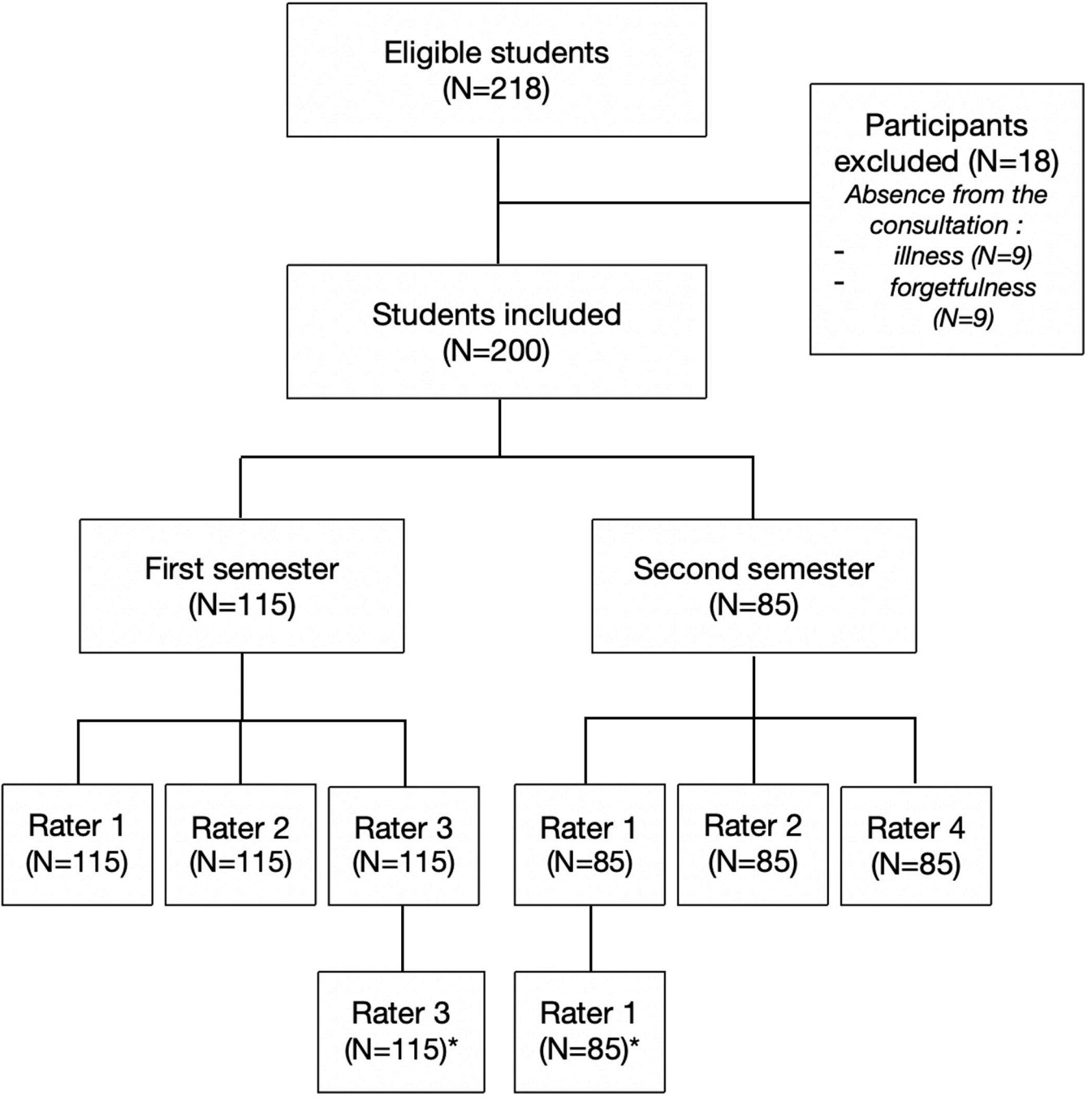

**Fig 1. Flow of medical students and raters throughout the study.**

cross-loadings over 0.30. Exploratory factor analysis of the French version of the 4-HCS resulted in the merging of three habits (namely, Invest in the beginning [all six items], Elicit the patient's perspective [two out of three items], and Demonstrate empathy [all four items]) in a single dimension.

As hypothesized, all mean habit scores were significantly higher for medical student consultations recorded at the second semester (Table 3). Most pairwise Pearson correlation

**Table 1. Summary statistics and internal consistency for the cross-cultural adaptation of the 4 Habit Coding Scheme into French (n = 200).**

| Habit [range]–Item | Mean score (SD) | | Ceiling effect, n (%) | | Floor effect, n (%) | | Average inter-item correlation | Item-total correlation | Item-rest correlation* | Cronbach Alpha† |
|---|---|---|---|---|---|---|---|---|---|---|
| **1. Invest in the beginning [6–30]** | 18.77 | (2.80) | 0 | (0) | 0 | (0) | 0.48 | - | - | 0.80 |
| 1. Show familiarity | 3.09 | (0.18) | 0 | (0) | 0 | (0) | 0.43 | 0.55 | 0.50 | 0.81 |
| 2. Greet warmly | 3.18 | (0.27) | 0 | (0) | 0 | (0) | 0.48 | 0.64 | 0.57 | 0.79 |
| 3. Engage in small talk | 2.66 | (0.89) | 6 | (3.0) | 4 | (2.0) | 0.46 | 0.79 | 0.62 | 0.76 |
| 4. Question style | 3.25 | (0.74) | 2 | (1.0) | 0 | (0) | 0.52 | 0.84 | 0.72 | 0.73 |
| 5. Expansion of concerns | 3.51 | (0.70) | 0 | (0) | 4 | (2.0) | 0.50 | 0.82 | 0.70 | 0.73 |
| 6. Elicit full agenda | 3.07 | (0.81) | 3 | (1.5) | 1 | (0.5) | 0.47 | 0.77 | 0.60 | 0.76 |
| **2. Elicit the patient's perspective [3–15]** | 10.01 | (2.06) | 0 | (0) | 0 | (0) | 0.48 | - | - | 0.72 |
| 7. Patient's understanding | 3.70 | (0.78) | 0 | (0) | 4 | (2.0) | 0.48 | 0.79 | 0.55 | 0.62 |
| 8. Goals for visit | 2.81 | (0.79) | 6 | (3.0) | 0 | (0) | 0.51 | 0.82 | 0.60 | 0.57 |
| 9. Impact on life | 3.49 | (0.98) | 2 | (1.0) | 14 | (7.0) | 0.43 | 0.81 | 0.49 | 0.72 |
| **3. Demonstrate empathy [4–20]** | 13.75 | (2.95) | 0 | (0) | 3 | (1.5) | 0.52 | - | - | 0.87 |
| 10. Encourage emotional expression | 3.33 | (0.90) | 1 | (0.5) | 9 | (4.5) | 0.61 | 0.90 | 0.84 | 0.81 |
| 11. Accept feelings | 3.63 | (0.73) | 1 | (0.5) | 4 | (2.0) | 0.61 | 0.87 | 0.81 | 0.82 |
| 12. Identify feelings | 2.60 | (1.04) | 26 | (13.0) | 23 | (1.5) | 0.55 | 0.84 | 0.73 | 0.84 |
| 13. Show good nonverbal behavior | 4.19 | (0.71) | 0 | (0) | 42 | (20.5) | 0.53 | 0.76 | 0.66 | 0.84 |
| **4. Invest in the end [10–50]** | 33.91 | (6.13) | 0 | (0) | 0 | (0) | 0.46 | - | - | 0.88 |
| 14. Use patient's frame of reference | 3.60 | (0.81) | 0 | (0) | 15 | (7.5) | 0.45 | 0.70 | 0.56 | 0.86 |
| 15. Allow time to absorb | 4.37 | (0.56) | 0 | (0) | 33 | (16.5) | 0.36 | 0.55 | 0.43 | 0.88 |
| 16. Give clear explanation | 4.24 | (0.76) | 0 | (0) | 63 | (31.5) | 0.12 | 0.29 | 0.15 | 0.90 |
| 17. Offer rationale for tests | 3.56 | (0.95) | 8 | (4.0) | 10 | (5.0) | 0.45 | 0.73 | 0.63 | 0.86 |
| 18. Test for comprehension | 3.33 | (0.96) | 8 | (4.0) | 4 | (2.0) | 0.59 | 0.89 | 0.84 | 0.84 |
| 19. Involve in decision | 3.01 | (0.83) | 7 | (3.5) | 2 | (1.0) | 0.57 | 0.86 | 0.81 | 0.84 |
| 20. Explore plan acceptability | 3.21 | (0.98) | 6 | (3.0) | 3 | (1.5) | 0.58 | 0.87 | 0.82 | 0.84 |
| 21. Explore barriers | 2.02 | (0.84) | 47 | (23.5) | 0 | (0) | 0.46 | 0.73 | 0.64 | 0.86 |
| 22. Encourage questions | 3.56 | (0.93) | 1 | (0.5) | 25 | (12.5) | 0.44 | 0.72 | 0.61 | 0.86 |
| 23. Plan for follow-up | 3.00 | (0.95) | 8 | (4.0) | 1 | (0.5) | 0.43 | 0.70 | 0.58 | 0.87 |
| Overall [23–115] | 76.44 | (12.34) | | | | | 0.42 | - | - | 0.94 |

Abbreviations: SD, standard deviation.

* Item-rest correlation was computed as the correlation between an item and the composite score that was formed by all other items in the habit.

† Item Cronbach alpha was computed for composite score that was formed by all other items in the habit.

coefficients between habit and physician–patient relationship competence assessment scale scores were higher than 0.70, indicating satisfactory convergent validity (Table 4).

None of the four habits fulfilled the 0.70 inter-rater reliability criterion, with individual absolute-agreement ICC point estimates ranging from 0.42 to 0.64 (Table 5). Interestingly, the average absolute-agreement ICC for the 4-HCS and three out of four habits fulfilled the inter-rater reliability criterion of 0.70.

The individual absolute-agreement ICC was 0.72 for intra-rater reliability of the 4-HCS score, with point estimates ranging from 0.45 to 0.71 across habits. The average absolute-agreement ICC for the 4-HCS and all habits fulfilled the 0.70 intra-rater reliability criterion.

**Table 2. Exploratory factor analysis for the 23 items of the cross-cultural adaptation of the 4-Habit Coding Scheme into French after orthogonal Varimax rotation (n = 200)\*.**

| Habit–Item | Factor 1 | Factor 2 | Factor 3 | Factor 4 |
|---|---|---|---|---|
| 1. Invest in the beginning | | | | |
| 1. Show familiarity | **.41** | | | .40 |
| 2. Greet warmly | **.40** | .34 | | .33 |
| 3. Engage in small talk | **.58** | | | |
| 4. Question style | **.58** | .49 | .32 | |
| 5. Expansion of concerns | **.74** | .33 | | |
| 6. Elicit full agenda | **.64** | .32 | | |
| 2. Elicit the patient's perspective | | | | |
| 7. Patient's understanding | .53 | **.61** | | |
| 8. Goals for visit | **.48** | .45 | .47 | .35 |
| 9. Impact on life | **.58** | | | |
| 3. Demonstrate empathy | | | | |
| 10. Encourage emotional expression | **.91** | | | |
| 11. Accept feelings | **.74** | 0.35 | | |
| 12. Identify feelings | **.86** | | | |
| 13. Show good nonverbal behavior | **.54** | .37 | | |
| 4. Invest in the end | | | | |
| 14. Use patient's frame of reference | .33 | **.73** | | |
| 15. Allow time to absorb | .36 | | | |
| 16. Give clear explanation | | | | |
| 17. Offer rationale for tests | | **.71** | | |
| 18. Test for comprehension | .31 | **.84** | .42 | |
| 19. Involve in decision | .39 | **.76** | .67 | |
| 20. Explore plan acceptability | | **.78** | .63 | |
| 21. Explore barriers | | **.61** | .46 | |
| 22. Encourage questions | | **.69** | | |
| 23. Plan for follow-up | | **.56** | .47 | |
| Overall variance explained, % | 46.5 | 8.8 | 6.7 | 4.8 |

\* Values are item loadings ≥.30

After restricting our analytical sample to the most experienced raters (raters #1 and #2), the individual absolute-agreement ICC was 0.83 for inter-rater reliability and 0.89 for intra-rater reliability.

**Table 3. Comparison of 4-Habit Coding Scheme scores for medical student consultations recorded during first and second semesters.**

| Habit [range] | 1st semester (n = 115) | | 2nd semester (n = 85) | | P |
|---|---|---|---|---|---|
| 1. Invest in the beginning [6–30] | 17.54 | (2.56) | 19.83 | (2.56) | < .001 |
| 2. Elicit the patient's perspective [3–15] | 8.88 | (1.93) | 10.98 | (1.63) | < .001 |
| 3. Demonstrate empathy [4–20] | 13.02 | (3.34) | 14.36 | (2.43) | < .001 |
| 4. Invest in the end [10–50] | 31.84 | (5.95) | 31.84 | (35.70) | < .001 |
| Overall [23–115] | 71.28 | (12.09) | 80.87 | (10.77) | < .001 |

\* Values are mean (standard deviation)

**Table 4. Correlation of the cross-cultural adaptation of the 4-Habit Coding Scheme into French and the physician–patient relationship competence assessment scale (n = 200)\*.**

| Habit [range] | Physician–patient relationship competence assessment scale | | Overall |
| --- | --- | --- | --- |
| | 1. Understanding of the patient's experience | 2. Communication, consultation management | |
| 1. Invest in the beginning | .85 | .61 | .81 |
| 2. Elicit the patient's perspective | .92 | .47 | .79 |
| 3. Demonstrate empathy | .77 | .64 | .76 |
| 4. Invest in the end | .70 | .79 | .78 |
| Overall | .90 | .74 | .90 |

\* Values are Pearson correlation coefficients. All *P*-values were < .001.

## Discussion

The use of validated standardized instruments is advocated for assessing physicians' interpersonal skills [8]. Indeed, validated instruments are likely to accurately reflect the concept to be measured while standardized instruments allow large-scale comparisons of physicians' communication skills across studies [30, 35]. The 4-HCS was therefore developed and validated from over 1,025 video-recorded medical consultations across various settings in the US and Western Europe (Table 6). Yet, our study was the first to examine the underlying factor structure of the 4-HCS and to report on its cross-cultural adaptation into French.

Each of the 200 video-recorded medical consultations in this study was rated without missing data by three out of four different raters, reflecting the acceptability and feasibility of the French version of the 4-HCS. Although the consultations were simulated with standardized patients and involved 4-year medical students in this study, the mean overall and subscale 4-HCS scores were consistent with previous reports (Table 6).

Surprisingly, the French version of the 4-HCS yields better performance regarding internal consistency than the original US instrument and previous cross-cultural adaptations. Cronbach's alpha was higher than 0.70 across habit subscales (median 0.83; range, 0.72–0.88) and compared favorably with those reported in the original US development (median, 0.66; range, 0.51–0.81) and German cross-cultural adaptation (median, 0.39; range, 0.31–0.46) (Table 6). This finding contrasts with previous studies that usually report worse performance for cross-cultural adaptations in comparison with original standardized instruments [30]. Krupat et al. were not concerned by the moderate internal consistency of the original version of the 4-HCS and speculated that successful training in communication would result in more effective communication and therefore greater internal consistency [8]. Another potential explanation for this discrepancy may be that the conceptual framework of the 4-HCS lacks generalizability or

**Table 5. Absolute-agreement intra-class correlation coefficient estimates for inter- and intra-rater reliability for the cross-cultural adaptation of the 4-Habit Coding Scheme into French (n = 200).**

| Habit | Inter-rater ICC (95% CI) | | | | Intra-rater ICC (95% CI) | | | |
| --- | --- | --- | --- | --- | --- | --- | --- | --- |
| | Individual | | Average | | Individual | | Average | |
| 1. Invest in the beginning | .45 | (.36 to .53) | .71 | (.63 to .77) | .59 | (.52 to .65) | .85 | (.81 to .88) |
| 2. Elicit the patient's perspective | .42 | (.33 to .60) | .68 | (.60 to .75) | .45 | (.37 to .52) | .76 | (.71 to .81) |
| 3. Demonstrate empathy | .53 | (.45 to .60) | .77 | (.71 to .82) | .58 | (.51 to .64) | .85 | (.81 to .88) |
| 4. Invest in the end | .64 | (.58 to .71) | .84 | (.80 to .88) | .71 | (.66 to .76) | .91 | (.89 to .93) |
| Overall | .60 | (.53 to .67) | .82 | (.77 to .86) | .72 | (.67 to .77) | .91 | (.89 to .93) |

Abbreviations: CI, confidence interval; ICC, intra-class correlation.

**Table 6. Primary studies reporting on the development or cross-cultural adaptation of the 4-Habit Coding Scheme.**

| Author, year | Krupat, 2006 | Fossli Jensen, 2010 | Clayton, 2011 | Scholl, 2014 | Present study |
|---|---|---|---|---|---|
| Country | USA | Norway | USA | Germany | France |
| Setting | Hospital | Hospital | Family practice clinics | Primary and specialty consultations | School of medicine |
| Recruitment period | 1994 | 2007–2008 | - | 2009–2010 | 2017–2018 |
| Experience | Resident and senior staff | Resident and senior staff | Resident and senior staff | Senior staff | Medical students |
| No. physicians | 50 | 71 | 21 | 22 | 200 |
| Simulated consultations | No | No | No | No | Yes |
| Recording | Video-recorded | Video-recorded | Video-recorded | Audio-taped | Video-recorded |
| No. consultations | 100 | 497 | 174 | 54 | 200 |
| Mean score | | | | | |
| 1. Invest in the beginning | 17.7 | - | 24.1 | 12.1 | 18.8 |
| 2. Elicit the patient's perspective | 7.6 | - | 11.5 | 4.5 | 10.0 |
| 3. Demonstrate empathy | 11.3 | - | 14.5 | - | 13.7 |
| 4. Invest in the end | 31.5 | - | 33.0 | 26.5 | 33.9 |
| Overall | 68.0 | 60.1 | 83.1 | - | 76.4 |
| Exploratory factor analysis | Not performed | Not performed | Not performed | Not performed | 4 principal components (66.8% of overall variance) |
| Cronbach's alpha | | | | | |
| 1. Invest in the beginning | .71 | - | - | .41 | .80 |
| 2. Elicit the patient's perspective | .51 | - | - | .46 | .72 |
| 3. Demonstrate empathy | .81 | - | - | .38 | .87 |
| 4. Invest in the end | .61 | - | - | .31 | .88 |
| Overall | - | .85 | - | - | .94 |
| Inter-rater reliability* | | | | | |
| 1. Invest in the beginning | .70 | - | .48 | .83 | .45 |
| 2. Elicit the patient's perspective | .80 | - | .57 | .79 | .42 |
| 3. Demonstrate empathy | .71 | - | .39 | .85 | .53 |
| 4. Invest in the end | .69 | - | .65 | .78 | .64 |
| Overall | .72 | .78 | .72 | - | .60 |
| Intra-rater reliability* | | | | | |
| 1. Invest in the beginning | - | - | - | .87 | .59 |
| Author, year | Krupat, 2006 | Fossli Jensen, 2010 | Clayton, 2011 | Scholl, 2014 | Present study |
| 2. Elicit the patient's perspective | - | - | - | .72 | .45 |
| 3. Demonstrate empathy | - | - | - | .84 | .58 |
| 4. Invest in the end | - | - | - | .83 | .71 |
| Overall | - | - | - | - | .72 |

\* Inter-rater reliability was quantified by the Pearson correlation coefficient in the studies by Krupat et al. and Clayton et al., and by the intra-class correlation coefficient in the study by Fossli Jensen et al.

‡ Inter- and intra-rater reliability scores were quantified by computing absolute agreement intra-class correlation coefficient in the study by Scholl et al. In the present study, individual absolute agreement intra-class correlation coefficient was used for assessing inter- and intra-rater reliability.

robustness and therefore does not apply equally to all to target populations' experience. Noticeably, the French version of the 4-HCS was used for rating medical student consultations

with standardized patients while the original US version was used for rating resident and senior staff medical consultations (Table 6).

To our knowledge, no previous study examined the factor structure of the 4-HCS. In confirmatory factor analysis, goodness-of-fit statistics did not support the hypothesized 4-dimension structure for the French version of the 4-HCS. Exploratory factor analysis resulted in two dimensions, with the merging of three conceptually related habits (Invest in the beginning, Elicit the patient's perspective, Demonstrate empathy) in a single dimension. Additionally, substantial cross-loading was observed for 15 out of 23 items, suggesting that the underlying factor structure of the French version of the 4-HCS was questionable.

These findings do not necessarily invalidate the postulated structure of the 4-HCS. Indeed, factor analysis can only discriminate uncorrelated constructs in a data set [36]. That three habits were correlated with each other in the present study sample does not imply that these scales measure the same concept [36]. The medical students participating in the current study were not trained with the Four Habit Model before their communication skills were assessed using the 4-HCS. This might explain why the factor structure for the French version departed from the postulated four-dimension structure of the 4-HCS. Yet, we cannot exclude that this issue is inherent to the original 4-HCS rather than being specific to our study sample. No previous study (including the original development study) examined the factor structure of the 4-HCS. Hence, there is a need for further investigation of the underlying factor structure of the original US version of the 4-HCS.

One item (16. *Give clear explanation*) was not allocated to any empirical dimension in exploratory factorial analysis and also yielded the lowest item-rest correlation, deteriorating the internal consistency of the corresponding habit scale. Altogether, these two observations question the relevance of this item, and therefore its removal from the French version of the 4-HCS should be discussed.

Evidence for convergent validity was provided by the expected correlation between the 4-HCS and physician–patient relationship competence assessment scale scores. Our observation that medical students more experienced in basic interview tasks yielded higher scores for all four habits supported the construct validity of the 4-HCS.

Comparisons of inter- and intra-reliability estimates were confounded by between-study heterogeneity in the types of correlation coefficients used. Only two studies, including the present one, used absolute-agreement ICC (Table 6). ICC is a recommended alternative to Pearson's r coefficient correlation for assessing inter- or intra-reliability [14, 31]. The median absolute-agreement ICC for inter-rater (0.49, range, 0.42–0.64) and intra-rater (0.58, range, 0.45–0.71) reliability for the French version of the 4-HCS were lower than those reported in the German cross-cultural adaptation study (0.81, range, 0.78–0.85 and 0.83, range, 0.72–0.87, respectively). Lower inter-rater reliability might be explained by the use of simulated consultations with standardized patients, the use of video- rather than audio-taped consultations, or varying levels of rater experience in our study. We observed that the greater the level of rater experience, the higher the 4-HCS scores in our study. Inconsistent associations have been reported between experience or seniority and communication scores in objective structured clinical examinations [37]. After restricting our analytical sample to the most experienced raters (i.e., raters #1 and #2), ICC estimates were higher than 0.80 for inter-rater reliability.

Interestingly, substantial improvement in inter-rater (median, 0.74; range, 0.68–0.84) and intra-rater (0.85, range, 0.76–0.91) reliability of habit scores was achieved using average absolute-agreement ICC estimates (Table 5). This latter finding supports the need for using two independent raters to rate communication skills with the 4-HCS based on video-recorded medical consultations, in routine practice.

This study has potential implications for routine assessment of physician communication skills using the cross-cultural adaptation of the 4-HCS into French. First, this study provides evidence on the validity of the 4-HCS scale for simulated consultations with standardized patients. Second, the French version of the 4-HCS scale demonstrated internal consistency, which was even higher than the original US version, allowing international comparisons. Third, our study questioned the hypothesized underlying factor structure of the 4-HCS. Since the 4-HCS was originally developed in the US, the conceptual framework and factor structure may lack generalizability or robustness and therefore may not apply equally well to other countries. Investigating the factor structure of the original US version is required to address this issue. Fourth, the moderate reliability of the French version of the 4-HCS implies that communication skills should be assessed by two independent experienced raters.

This cross-cultural adaptation study has a few caveats that must be considered. First, real patient encounters would have been preferable to standardized patient encounters for assessing communication skills, because of their authenticity [38]. Indeed, simulated consultations with standardized patients differ from real patient consultations in many ways [39]. Simulated patients are not suffering from illness and only attempt to portray the same through their acting. Moreover, recruiting and training standardized patients is time consuming in order to produce a high-quality simulation [40]. Although our study did not explore real patient encounters, standardized patients allowed us to provide a large number of students with reproducible and consistent clinical scenarios of the same level of difficulty [41].

Second, our study was conducted with 4th-year medical students at a single university-affiliated hospital and the findings may not apply to other settings. A broader spectrum of participants would strengthen the confidence in the psychometric properties of this cross-cultural adaptation of the 4-HCS into French.

Third, the 4-HCS and physician–patient relationship competence assessment scale were completed by the same rater so that a halo effect cannot be excluded when assessing convergent validity of the two instruments. Demonstrating that 4-HCS scores correlate with (standardized) patient-reported experience of physician communication skills would provide stronger evidence of construct validity.

Fourth, the one-semester interval separating the two groups of students might be too short to assess the relationship between 4-HCS scores and experience in communication and basic interview tasks, although the associations were significant. The ability of the 4-HCS to discriminate subjects with varying levels of communication skills warrants further investigation. The sensitivity of the French version of the 4-HCS to changes in communication skills also remains to be documented by way of longitudinal studies.

## Conclusions

The French version of the 4-HCS demonstrates satisfactory internal consistency but moderate reliability, requiring the use of two independent raters to assess communication skills of medical students based on video-recorded consultations with standardized patients. The empirical factor structure of the French version does not conform with the hypothesized habits of the original 4-HCS. Whether this reflects a specific issue with our cross-cultural adaptation study sample or a more general problem with the instrument is unclear and deserves further investigation.

## Supporting information

**S1 Appendix. Cross-cultural adaptation of the 4-HCS scale into French.**
(DOCX)

**S2 Appendix. Scree plot of the 23 items of the 4-HCS scale.**
(TIF)

# Acknowledgments

The authors are indebted to Pauline Bouchet and Ariane Martinez, Grenoble Alpes University, for their collaboration in setting up simulated medical consultations with standardized patients, Noémie Kaladzé and Alexis Dechosal, Grenoble Alpes University, for rating video-recorded medical consultations. The authors thank Alison Foote, Grenoble Alpes University Hospital, and Linda Northrup for their help in editing the manuscript in English.

# Author Contributions

**Conceptualization:** Alexandre Bellier, Philippe Chaffanjon, Edward Krupat, Patrice Francois, José Labarère.

**Data curation:** Alexandre Bellier.

**Formal analysis:** Alexandre Bellier, José Labarère.

**Funding acquisition:** Alexandre Bellier.

**Investigation:** Alexandre Bellier, Philippe Chaffanjon.

**Methodology:** Alexandre Bellier, José Labarère.

**Project administration:** Alexandre Bellier, Philippe Chaffanjon, Patrice Francois, José Labarère.

**Resources:** Edward Krupat.

**Software:** Alexandre Bellier, José Labarère.

**Supervision:** Philippe Chaffanjon, Patrice Francois, José Labarère.

**Validation:** Philippe Chaffanjon, José Labarère.

**Visualization:** Philippe Chaffanjon, Patrice Francois.

**Writing – original draft:** Alexandre Bellier.

**Writing – review & editing:** Philippe Chaffanjon, José Labarère.

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
