## [Decision Letter · Decision Letter 0]

22 Oct 2019

PONE-D-19-24879

Cross-cultural adaptation of the 4-Habits Coding Scheme into French to assess physician communication skills

PLOS ONE

Dear Dr. BELLIER,

Thank you for submitting your manuscript to PLOS ONE. After careful consideration, we feel that it has merit but does not fully meet PLOS ONE’s publication criteria as it currently stands. Therefore, we invite you to submit a revised version of the manuscript that addresses the points raised during the review process.

We would appreciate receiving your revised manuscript by Dec 06 2019 11:59PM. To enhance the reproducibility of your results, we recommend that if applicable you deposit your laboratory protocols in protocols.io, where a protocol can be assigned its own identifier (DOI) such that it can be cited independently in the future. For instructions see: http://journals.plos.org/plosone/s/submission-guidelines#loc-laboratory-protocols

We look forward to receiving your revised manuscript.

Kind regards,

Gian Mauro Manzoni, Ph.D., Psy.D.

Academic Editor

PLOS ONE

Journal Requirements:

1.

Reviewers' comments:

Reviewer's Responses to Questions

**Comments to the Author**

1. Is the manuscript technically sound, and do the data support the conclusions?

Reviewer #1: Partly

Reviewer #2: Yes

2. Has the statistical analysis been performed appropriately and rigorously? 

Reviewer #1: Yes

Reviewer #2: Yes

3. Have the authors made all data underlying the findings in their manuscript fully available?

Reviewer #1: Yes

Reviewer #2: Yes

4. Is the manuscript presented in an intelligible fashion and written in standard English?

Reviewer #1: Yes

Reviewer #2: Yes

5. Review Comments to the Author

Reviewer #1: This brief manuscript reports upon a French-language validation of an instrument for assessing physicians’ communication to patients (4-Habits Coding Scheme, 4HCS). The findings and the previous literature are clearly laid out in easily digested formats (e.g. informative tables). I only have three substantive concerns.

1. Unlike previous work with this coding scheme, the authors rightly use ICC to assess interrater reliability. However, it is problematic that those ICC reliability scores are so low. The low reliability implies that you may not be accurately measuring physician communication. Can you estimate the amount of noise in your ratings? That is, how much of the variance in the four habits might be due to inconsistent measurement? Also, I would like to hear more about the differences between raters, including greater contextualization of the ratings in the literature.

2. Actual patient interactions differ from enacted consultations in many important ways. The examination of simulated interactions in the current study is not necessarily a problem, but it is distinct. Thus, the research report should more fully explore how such an application of 4HCS contrasts with a genuine doctor-patient meeting.

3. Kaiser developed the four habits to guide physicians’ interactions with patients. Then, researchers assessed their communication with the 4HCS, studying the extent to which they employed those target strategies. How similar is that original context in which the 4HCS was developed to the medical school where your data were collected? To what extent does the curriculum include the four habits? I ask this question in light of the factor analysis finding only two, rather than the four, dimensions of the 4HCS. One might not hypothesize finding all four habits in a population where physician training differs substantially from the original setting.

4. Some minor suggestions. Line 59 “supporting that” is awkward. Try “supporting the conclusion that” or “suggesting that.” Lines 214-5: double negative maybe unavoidable here, but you could add a line to increase clarity. Same suggestion for line 290. Lines 219-220: Are these thresholds common? Cite someone to support their appropriateness.

Best of luck on your manuscript.

Reviewer #2: Thank you for allowing me to review your article. I find the manuscript overall in decent shape. The objectives and aims of your study are clear. The study is itself useful. I did have several issues and questions I summarize below that led me to recommend that you revise and resubmit.

1. Provide evidence for the cultural (in)variance of the 4-HCS. The tool has been translated into several languages. Can you briefly summarize the psychometric findings of these instruments (e.g., what were CFA results in these studies)? In yet other words, would you expect, on the basis of these past studies, to reproduce the factor structure that was generated with the US American samples?

2. What *is* the potential of the 4-HCS?

3. There is arguably a problematic issue with the 4-HCS: it does not help physicians learn communication skills, which are behaviors. The tool merely assesses whether trained observers can identify four stages of the physician-patient conversation. HOW these stages are enacted is neither coded, nor assessed; nor does the tool assess how "well* the physician, say, demonstrates empathy; that is what does "good" nonverbal behavior mean? Indeed, the generality and broadness of the scale is what makes it useful. I would certainly not use it to train communication competency.

4. What was the length of the standardized patient conversations?

5. I don't quite understand why you conducted the CFA first? It seems to me that we usually run a structural model first before we run a measurement model? It seems to me that if you are predicting four factors, you want to use CFA procedures. You may not need the EFA. But perhaps I am not quite understanding your analyses. I admit, I had a bit of a hard time following your data write-up since it's all not quite in APA style.

6. I am missing a discussion of the practical implications. So what about cross-cultural differences in US and French samples for the measures? What does that mean for physicians and patients and communication skills.

7. In all, the manuscript could probably profit from a really strong edit.

Thank you for letting me review this manuscript. I'd be delighted to review it once, should the editor extend a revise and resubmit.

6. PLOS authors have the option to publish the peer review history of their article (what does this mean?). If published, this will include your full peer review and any attached files.

Reviewer #1: No

Reviewer #2: No

---

## [Author Response · Author response to Decision Letter 0]

5 Dec 2019

Reviewer #1

This brief manuscript reports upon a French-language validation of an instrument for assessing physicians’ communication to patients (4-Habits Coding Scheme, 4HCS). The findings and the previous literature are clearly laid out in easily digested formats (e.g. informative tables). I only have three substantive concerns.

We are grateful to this reviewer for his/her kind assessment of our study.

1. Unlike previous work with this coding scheme, the authors rightly use ICC to assess interrater reliability. However, it is problematic that those ICC reliability scores are so low. The low reliability implies that you may not be accurately measuring physician communication. Can you estimate the amount of noise in your ratings? That is, how much of the variance in the four habits might be due to inconsistent measurement? Also, I would like to hear more about the differences between raters, including greater contextualization of the ratings in the literature.

We agree with this reviewer that ICC estimates were disappointingly low for interrater reliability in our study. This observation might reflect heterogeneity in the level of clinical experience across raters. Consistent with Krupat et al., we recruited senior physicians as well as medical students for rating interpersonal skills using the 4-HCS. A significant trend towards higher scores was found as the level of clinical experience increased, with mean overall scores ranging from 66.97 (SD 10.26) to 93.42 (SD 13.46) out of 115 for rater #4 (a medical student) and rater #1 (a senior physician), respectively. Noticeably, this trend was the opposite of that reported in a previous study (see Chong et al. J Educ Eval Health Prof 2018.). After restricting our analytical sample to the most experienced raters (raters #1 and #2), ICC estimates were higher than 0.80 for inter-rater reliability. 

To address this reviewer’s comment, we incorporated this new information in the Results and Discussion sections of the revised manuscript (page 20, line 340 and page 24, line 427). 

2. Actual patient interactions differ from enacted consultations in many important ways. The examination of simulated interactions in the current study is not necessarily a problem, but it is distinct. Thus, the research report should more fully explore how such an application of 4HCS contrasts with a genuine doctor-patient meeting.

The Reviewer is right that actual physician–patient interactions differ from consultations with standardized patients in many ways (Cleland et al. 2009). 

As suggested, we elaborated on this point in the Discussion section of the revised manuscript (page 22, line 371). 

3. Kaiser developed the four habits to guide physicians’ interactions with patients. Then, researchers assessed their communication with the 4HCS, studying the extent to which they employed those target strategies. How similar is that original context in which the 4HCS was developed to the medical school where your data were collected? To what extent does the curriculum include the four habits? I ask this question in light of the factor analysis finding only two, rather than the four, dimensions of the 4HCS. One might not hypothesize finding all four habits in a population where physician training differs substantially from the original setting.

This reviewer is right that the 4 Habits Coding Scheme (4-HCS) is based on the conceptual framework of the Four Habits Model, a training program that was developed for teaching effective communication to clinicians. We acknowledge that the medical students participating in our study were not trained with the Four Habits Model before their communication skills were assessed using the 4-HCS. This might explain why the factor structure for the French version departed from the hypothesized 4-dimension structure of the 4-HCS. However, it should be underlined that no previous study (including the original development study of this scale) has examined the factor structure of the 4-HCS study. A paragraph addressing this issue was added to the Discussion section of the revised manuscript (page 23, line 400).

4. Some minor suggestions. Line 59 “supporting that” is awkward. Try “supporting the conclusion that” or “suggesting that.” Lines 214-5: double negative maybe unavoidable here, but you could add a line to increase clarity. Same suggestion for line 290.

We have made the proposed corrections (page 4, line 59 and page 11, line 216).

5. Lines 219-220: Are these thresholds common? Cite someone to support their appropriateness.

The following reference supporting the threshold used for Varimax rotation was incorporated into the list of citations of the revised manuscript (page 11, line 221).

 

Reviewer #2

Thank you for allowing me to review your article. I find the manuscript overall in decent shape. The objectives and aims of your study are clear. The study is itself useful. I did have several issues and questions I summarize below that led me to recommend that you revise and resubmit.

We are grateful to this Reviewer for his/her kind assessment of our study.

1. Provide evidence for the cultural (in)variance of the 4-HCS. The tool has been translated into several languages. Can you briefly summarize the psychometric findings of these instruments (e.g., what were CFA results in these studies)? In yet other words, would you expect, on the basis of these past studies, to reproduce the factor structure that was generated with the US American samples?

Since the publication of the original US version, the 4-HCS has been translated into different languages, including Norwegian, German, and Brazilian Portuguese, and used across various settings. To our knowledge, only three studies (including the US development study) reported on the psychometric properties of the 4-HCS and none examined the underlying factor structure. The 4-HCS developers recommended further examining its validity and reliability in different settings and populations (Krupat E, et al. Patient Educ Couns 2006)

To address this reviewer’s concern, we summarized the psychometric properties for the original US version and the cross-cultural adaptations of the 4-HCS in Table 6. We also emphasized that our study was the first to examine the factor structure of the 4-HCS (page 5, line 83 and page 23, line 405).

2. What *is* the potential of the 4-HCS?

Because the 4-HCS scale is a standardized, validated, and reliable instrument, it has the potential for assessing baseline physician communication skills and/or measuring the effectiveness of a training program in altering physician communication skills. Importantly, the 4-HCS overcomes the challenges of unstructured assessment of physician–patient interaction during medical consultations, which is a complex task because of less tangible aspects of communication, often referred to as interpersonal skills (e.g., expression of humanistic attitudes such as understanding, supportiveness, and empathy). As suggested, we clarified this expression in the rationale of the revised manuscript (Page 4, line 80). 

3. There is arguably a problematic issue with the 4-HCS: it does not help physicians learn communication skills, which are behaviors. The tool merely assesses whether trained observers can identify four stages of the physician-patient conversation. HOW these stages are enacted is neither coded, nor assessed; nor does the tool assess how "well* the physician, say, demonstrates empathy; that is what does "good" nonverbal behavior mean? Indeed, the generality and broadness of the scale is what makes it useful. I would certainly not use it to train communication competency.

Actually, the 4-HCS scale is an assessment tool but not a training program by itself. The 4-HCS scale provides physicians with feedback on their own performance regarding basic medical interview tasks but does not help acquire communication skills. A training program remains to be developed and implemented for teaching effective communication skills.

This reviewer might have misunderstood that each item of the 4-HCS is rated on a 5-point Likert scale, ranging from 1 to 5, with higher scores denoting better performance. In the original US version, the midpoint (i.e., 3) and the two endpoints (i.e., 1 and 5) were anchored, with specific behavioral descriptions. To address this reviewer’s concern, we reported the translation of specific behavioral descriptions into French, in Appendix #1.

4. What was the length of the standardized patient conversations?

The median duration was 8 min (range, 4–19 min) per video-recorded consultation. This information was provided in the Results section of the original version of our manuscript (page 13, line 264).

5. I don't quite understand why you conducted the CFA first? It seems to me that we usually run a structural model first before we run a measurement model? It seems to me that if you are predicting four factors, you want to use CFA procedures. You may not need the EFA. But perhaps I am not quite understanding your analyses. I admit, I had a bit of a hard time following your data write-up since it's all not quite in APA style.

According to current guidelines addressing the cross-cultural adaptation of standardized instruments (Nunnaly and Bernstein. 1994), we first performed structural equation modeling of the 23 items assigned to the four intended habits to determine whether our data fit with the hypothesized internal structure of the original US version of the 4-HCS. Then we performed exploratory factor analysis in order to examine possible alternative structures to the original US version. The following reference supporting our CFA and EFA strategy was incorporated into the list of citations of the revised manuscript (page 10, line 205 and page 11, line 221).

6. I am missing a discussion of the practical implications. So what about cross-cultural differences in US and French samples for the measures? What does that mean for physicians and patients and communication skills?

Our study has potential implications for routine assessment of physician communication skills using the cross-cultural adaptation of the 4-HCS into French. First, our study provides evidence on the validity of 4-HCS scale for assessing communication skills based on simulated consultations with standardized patients. Second, the French version of the 4-HCS scale demonstrated internal consistency, which was even higher than the original US version, allowing large-scale international comparisons of study results. Third, our study questioned the hypothesized underlying factor structure of the 4-HCS. Because the 4-HCS was originally developed in the US, the conceptual framework and factor structure may lack generalizability or robustness and therefore may not apply equally well to other countries. Investigating the factor structure of the original US version is required to address this issue. Fourth, the moderate reliability of the French version of the 4-HCS implies that communication skills should be assessed by two independent experienced raters. As requested, a paragraph was added summarizing the practical implications of our study in the Discussion section of the revised manuscript (page 25, line 438).

7. In all, the manuscript could probably profit from a really strong edit.

As suggested, the revised manuscript has been edited by a native English speaker.

---

## [Decision Letter · Decision Letter 1]

16 Jan 2020

PONE-D-19-24879R1

Cross-cultural adaptation of the 4-Habits Coding Scheme into French to assess physician communication skills

PLOS ONE

Dear Dr. BELLIER,

Thank you for submitting your manuscript to PLOS ONE. After careful consideration, we feel that it has merit but does not fully meet PLOS ONE’s publication criteria as it currently stands. Therefore, we invite you to submit a revised version of the manuscript that addresses the points raised during the review process.

We would appreciate receiving your revised manuscript by Mar 01 2020 11:59PM. To enhance the reproducibility of your results, we recommend that if applicable you deposit your laboratory protocols in protocols.io, where a protocol can be assigned its own identifier (DOI) such that it can be cited independently in the future. For instructions see: http://journals.plos.org/plosone/s/submission-guidelines#loc-laboratory-protocols

We look forward to receiving your revised manuscript.

Kind regards,

Gian Mauro Manzoni, Ph.D., Psy.D.

Academic Editor

PLOS ONE

Reviewers' comments:

Reviewer's Responses to Questions

**Comments to the Author**

1. If the authors have adequately addressed your comments raised in a previous round of review and you feel that this manuscript is now acceptable for publication, you may indicate that here to bypass the “Comments to the Author” section, enter your conflict of interest statement in the “Confidential to Editor” section, and submit your "Accept" recommendation.

Reviewer #1: (No Response)

Reviewer #2: All comments have been addressed

2. Is the manuscript technically sound, and do the data support the conclusions?

Reviewer #1: Yes

Reviewer #2: Yes

3. Has the statistical analysis been performed appropriately and rigorously? 

Reviewer #1: Yes

Reviewer #2: Yes

4. Have the authors made all data underlying the findings in their manuscript fully available?

Reviewer #1: Yes

Reviewer #2: Yes

5. Is the manuscript presented in an intelligible fashion and written in standard English?

Reviewer #1: Yes

Reviewer #2: Yes

6. Review Comments to the Author

Reviewer #1: Well done!

For my comment 2 (Actual patient interactions differ from enacted consultations) on the previous submission, the revision doesn’t address my concern, and I’m afraid that my comment was unclear. Let me explain what I meant. As an expert in interpersonal communication, I can say that scholars who study health communication will likely be interested in your research. It would help you reach these readers if you gave a brief assessment of your choice of methods in this study of physicians’ communication skills. In particular, I’m suggesting that the authors clearly differentiate their work from a study with normal patients. There are clear advantages of using standardized patients, but there are also limitations, which remain unmentioned in the revision. So, I’m requesting a sentence or two to justify the choice to study enacted as opposed to genuine interactions. Perhaps something along the lines of “Although the study did not explore actual patient-doctor interaction, standardized patients allowed us to...”

Reviewer #2: Admittedly, I had several critical questions in my initial review of the manuscript. In my view, the authors have done a *most* effective job in addressing not only my concerns but the other reviewer's concerns as well. I am wholly satisfied with these changes.

7. PLOS authors have the option to publish the peer review history of their article (what does this mean?). If published, this will include your full peer review and any attached files.

Reviewer #1: No

Reviewer #2: No

---

## [Author Response · Author response to Decision Letter 1]

27 Jan 2020

Reviewer #1

For my comment 2 (Actual patient interactions differ from enacted consultations) on the previous submission, the revision doesn’t address my concern, and I’m afraid that my comment was unclear. Let me explain what I meant. As an expert in interpersonal communication, I can say that scholars who study health communication will likely be interested in your research. It would help you reach these readers if you gave a brief assessment of your choice of methods in this study of physicians’ communication skills. In particular, I’m suggesting that the authors clearly differentiate their work from a study with normal patients. There are clear advantages of using standardized patients, but there are also limitations, which remain unmentioned in the revision. So, I’m requesting a sentence or two to justify the choice to study enacted as opposed to genuine interactions. Perhaps something along the lines of “Although the study did not explore actual patient-doctor interaction, standardized patients allowed us to...”

We apologize for our failure in addressing this Reviewer’s comment adequately in the first revision of our manuscript. We agree with this reviewer that real patient encounters differ from consultations with standardized patients in many ways. As suggested, we have incorporated a paragraph addressing this important limitation of our study in the discussion of the revised manuscript (pages 25-26, lines 455-465). The revised section reads as follows:

“First, real patient encounters would have been preferable to standardized patient encounters for assessing communication skills, because of their authenticity [38]. Indeed, simulated consultations with standardized patients differ from real patient consultations in many ways [39]. Simulated patients are not suffering from illness and only attempt to portray the same through their acting. Moreover, recruiting and training standardized patients is time consuming in order to produce a high-quality simulation [40]. Although our study did not explore real patient encounters, standardized patients allowed us to provide a large number of students with reproducible and consistent clinical scenarios of the same level of difficulty [41].” 

Reviewer #2

Admittedly, I had several critical questions in my initial review of the manuscript. In my view, the authors have done a *most* effective job in addressing not only my concerns but the other reviewer's concerns as well. I am wholly satisfied with these changes.

We are grateful to this Reviewer for his/her kind assessment of our manuscript.

---

## [Decision Letter · Decision Letter 2]

6 Mar 2020

Cross-cultural adaptation of the 4-Habits Coding Scheme into French to assess physician communication skills

PONE-D-19-24879R2

Dear Dr. BELLIER,

We are pleased to inform you that your manuscript has been judged scientifically suitable for publication and will be formally accepted for publication once it complies with all outstanding technical requirements.

With kind regards,

Gian Mauro Manzoni, Ph.D., Psy.D.

Academic Editor

PLOS ONE

---

## [Editor Report · Acceptance letter]

25 Mar 2020

PONE-D-19-24879R2 

Cross-cultural adaptation of the 4-Habits Coding Scheme into French to assess physician communication skills 

Dear Dr. Bellier:

I am pleased to inform you that your manuscript has been deemed suitable for publication in PLOS ONE. Congratulations! Your manuscript is now with our production department. 

With kind regards,

on behalf of

Dr. Gian Mauro Manzoni 

Academic Editor

PLOS ONE